# A Case Study: Layout Optimization of Three Gorges Wind Farm Pakistan, Using Genetic Algorithm

**Muhammad Bin Ali [1], Zeshan Ahmad [1], Saad Alshahrani [2,\*], Muhammad Rizwan Younis [1], Irsa Talib [1] and Muhammad Imran [3]**

1. Department of Mechanical Engineering, School of Engineering (SEN), University of Management & Technology, C II Johar Town, Lahore 54770, Pakistan
2. Department of Mechanical Engineering, College of Engineering, King Khalid University, P.O. Box 394, Abha 61421, Saudi Arabia
3. Department of Mechanical, Biomedical and Design Engineering, College of Engineering and Physical Sciences, Aston University, Birmingham B4 7ET, UK
* Correspondence: saadayed@kku.edu.sa

**Abstract:** Wind is an important renewable energy source. The majority of wind farms in Pakistan are installed in Jhimpir, Sindh Wind Corridor. At this location, downstream turbines encounter upstream turbines', wake, decreasing power output. To maximize the power output, there is a need to minimize these wakes. In this research, a method is proposed to maximize the power output using a Genetic Algorithm (GA). Hub heights and inter-turbine spacing are considered variables in this method. Two wind farms located at Jhimpir, Sindh, namely, Second and Third Three Gorges Wind Farms (TGWFs), have been analyzed. Three different cases are considered to maximize the power output. In Case 1, thesame hub heights and inter-turbine spacing without wake effects are considered. In Case 2, the same hub heights and inter-turbine spacing with wake effects are considered. In Case 3, variable hub heights and inter-turbine spacing with wake effects are considered. The results revealed that TGWFs, with variable hub heights and inter-turbine spacing, produce more power output. It is also revealed that the increase in power output, in the case of two different hub heights, is greater in comparison to three different hub heights. Eventually, the proposed method may help in the layout optimization of a wind farm.

**Keywords:** wind energy; turbine; genetic algorithm; layout optimization; inter-turbine spacing; hub height

## 1. Introduction

Fossil fuels are continuously adding to global warming and pollution on earth. Nowadays, the temperature has been higher than at any time in the past 400,000 years [1]. To reduce global warming, an alternative energy resource like the wind is needed. The wake effect reduces the practical use of wind energy. In general, the wake effect is divided into two parts. The first part, immediately behind the turbine, is termed near wake. It continues up to the distance of two–five wind turbine rotor diameters (D). The second part beyond 5D is called a far wake region [2]. The wake effect significantly reduces the power output of wind turb inesas the wind turbines upstream take some of the wind's kinetic energy for themselves. It leads to a lower wind speed for the downstream wind turbines. In many wind farms, the negative impact of the wake on power out putis prominent. At Horns Rev wind farm, Denmark, a three-month power drop due to the wake effect is 21.6% [3]. At Yeongheung wind farm, Korea, annual energy production (AEP) reduces by 7% due to the wake effect [4]. Further, the AEP at Roscoe wind farm, Texas, reduces by 8% due to the wake effect [5]. The mean wind speed deficit due to the wake effect at Nysted wind farm in the Baltic Sea and Horns Rev wind farm in the North Sea is observed using Satellite Synthetic Aperture Radar. The mean wind speed deficit due to the wake effect tis 8–9% [6].

Wake models help to avoid the regions where the wake effect is maximum. In general, the wake models are divided into two categories: analytical and computational. Among these models, the Jensen wake model is the most primitive. It is also known as PARK Model. Later, other wake models, such as Frandsen Model and Larsen Model, were developed [7,8]. Shakoor et al. [9] have done a review of different wake models. Particular emphasis has been laid on the far-wake models in their work. These models ignored the regions close to the turbines because of high turbulence. It has been found in their work that far-wake models perform well in comparison to near-wake models.

Wind farm layout optimization is vital to minimize wake effects. It installs wind turbines at optimal positions where the wake effect is minimum. It has been observed that wake losses increase with the reduction in inter-turbine spacing [10]. There are design constraints in wind farm layout optimization. Researchers have formulated different methods to evade these constrained regions. One proposal is based on grid spacing [11]. Wind farm terrain also influences wind farm layout optimization. Terrain involves the natural features of a particular area. Han et al. [12] have applied the quadratic interpolation method to optimize a difficult terrain wind farm. This work has found that terrain optimization increases the wind speed, as well as the power output of a wind farm. Terrains in many regions have escarpments. Escarpments are steep slopes or long cliffs that separate two areas into different elevations. A slight change in shape, such as replacing the round edge of the escarpment with a sharp edge, can reduce the power output of wind farm sbyup to 50% [13]. An appropriate algorithm for wind farm layout optimization is essential. Marmidis et al. [14] have applied the MonteCarlo simulation algorithm for wind farm layout optimization. Particle swarm optimization (PSO) algorithm and evolutionary algorithm are applied by Shin et al. [15] to optimize the layout of a wind farm along the coast of Busan, South Korea. Another algorithm, a greedy algorithm, is applied by Kai et al. [16] for wind farm layout optimization. The greedy algorithm chooses the best option at the moment and does not carry out rankings of solutions. The greedy algorithm and genetic algorithm (GA) were compared by Elkinton et al. [17]. More precise measurements were received by GA. The algorithm for wind farm layout optimization in the current work is GA. In the majority of literatureon wind farm layout optimization, the preferred algorithm alsore mains GA. GA is a heuristic algorithm and follows Charles Darwin's theory of human reproduction. In wind technology, Mosetti et al. [18] were the first who applied GA to optimize the wind farm layout. GA was used along with the Jensen model. Later, Grady et al. [19] took guidance from Mosetti et al.'s work. Changes were brought in Mosetti et al.'s work, and the number of individuals and generations was enhanced. This increased the power output of wind farms. Mittal [20] furthered the work on GA for wind farm layout optimization and used the micro-sitting method to optimize the wind farm layout to enhance the power output. Bossi et al. [21] used 13 different genetic algorithms to optimize wind farm shapes. One was traditional GA, another was new GA, and the rest were hybridized GA. Khanali et al. [22] have used GA to increase the power output of the Tehran wind site, Iran. Three different scenarios are considered, and the one based on optimal longitudinal and latitude distances provided the optimal results. Similarly, Park et al. [23] have used GA to increase the power output of the Dwange long wind farmin South Korea. The power output was enhanced by 2.5%.

Hub height refers to the distance of the platform of a wind turbine from its rotor. Variation in hub heights reduces the impact of upstream wind turbines' wake on the downstream wind turbines. Ying et al. [24] first attempted to optimize wind farm hub heights using a genetic algorithm. It was done in three different scenarios: constant wind speed and direction, variable wind speed and constant wind direction, and variable wind speed and wind direction. Dupont et al. [25] also attempted to optimize wind farm hub heights using a multi-level extended three-pattern search algorithm. Wang et al. [26] applied different wake models on a wind farm with multiple hub heights. Wake models used by the authors were PARK Model, Larsen Model, and B–P Model. A comparison of the wake models was made. PARK Model and Larsen Model were found to be the most

efficient models. Vasel-Be-Hagh et al. [27] applied a different greedy algorithm to optimize the hub heights of wind turbines. The greedy algorithm was applied to the Lillgrund wind farmin Sweden, which produced 2% more AEP than average using different hub heights. In Pakistan, most wind farms are installed in Sindh Wind Corridor, which has a vast potential for wind energy generation. The research carried out by Saeed et al. [28] has analyzed eighteen potential wind energy sites in Pakistan and found six wind energy sites to be suitable. These sites are located in Sindh Wind Corridor. Similarly, Ahmed et al. [29] examined Pakistan's potential wind energy sites. Four potential sites—Karachi, Ormara, Pasni, and Gwadar—were identified for wind energy exploitation in Pakistan. It was found that there gion near Karachi was most suitable for wind energy generation in Pakistan. Khahro et al. [30] carried out a feasibility study of the Gharo site in the Sindh Wind Corridor. By evaluating the wind speed and direction for five years, it was concluded that this site is suitable for wind energy generation. Baloch et al. [31] also analyzed wind energy potential in the Sindh and Baluchistan provinces of Pakistan. It was found that the area around Jhimpir, Sindh, is suitable for wind energy exploitation in Pakistan. The precise and supportive leadership is now required to make wind power a success in Pakistan [32]. In fact, at Jhimpir, Sindh, most of the wind farms suffer from the wake effects. Not much work has been done to examine wake effects to date. Consequently, the need for research to enhance Jhimpir wind farms' power output through layout optimization has beenen hanced.

In this work, the methodology used to optimize the wind farm layout to maximize the power output has been discussed in Section two. Firstly, problem formulation has been done. Secondly, the Jensen wake model has been described, and a brief description of the genetic algorithm (GA) has been given. In the next section, the second and third three Gorges Wind Farms (TGWFs) on which GA is applied are discussed. In Section four, results and discussion have been done. Three different cases have been considered: Case 1, same hub height and inter-turbine spacing without wake effect; Case 2, same hub height and inter-turbine spacing with wake effect, and Case 3, variable hub height and inter-turbine spacing with wake effect. Case 3 is further divided into two subcases: (i) two different hub heights and variable inter-turbine spacing, (ii) three different hub heights, and variable inter-turbine spacing.

## 2. Materials and Methods

### 2.1. Problem Formulation

The wind farm layout optimization problem is defined as follows: to find the optimal layout of Three Gorges Wind Farms (TGWFs) to maximize the power output by varying the hub heights and inter-turbine spacing, using Genetic Algorithm (GA) while considering the wake effect. Power output is calculated using Equation (1), and constraints are defined using Equations (2) and (3).

$$Maximize P = \sum_{i=1}^{n} p_i \tag{1}$$

$$Subject\ to\ H_{min} \leq H \leq H_{max} \tag{2}$$

$$S_{min} \leq S \leq S_{max} \tag{3}$$

$$i = 1, 2, 3, \ldots, n$$

where,

$P$: Total power output
$p_i P$: Power output of the individual turbine
$n$: Number of turbines
$H$: Hub height
$S$: Inter-turbine spacing
*min* and *max*: Lower and upper limits of hub heights and inter-turbine spacing.

### 2.2. Jensen Wake Model

The Jensen wake model is a far-wake model and was further developed by Katic et al. [33,34]. The boundary condition of the Jensen wake model is to neglect the near-turbulence intensity. Angular momentum is considered conserved inside the wake. No external forces act upon the control volume, as shown in Figure 1. The probability is that the wake effect expands linearly along the 'x' direction, allowing the Jensen wake model to operate more efficiently than the other wake models [35].

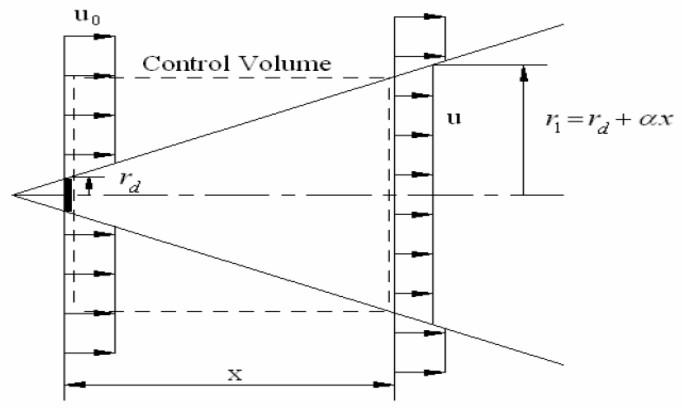

**Figure 1.** Schematic diagram of the Jensen wake model [34].

Power output is calculated using Equation (4).

$$p_i = \eta \frac{1}{2} \rho A u_i^{\ 3} \tag{4}$$

where,

$p_i$: Power output of the individual turbine
$\eta$: Betz Constant
$\rho$: Density
$A$: Cross-sectional area of turbine
$u_i$: Wind velocity

In Equation (4), '$\eta$' is Betz Constant given by Betz and it cannot exceed 0.59 [36]. According to the Jensen model, wake radius '$r_w$' is calculated using Equation (5).

$$r_w = r_d + \alpha x \tag{5}$$

where,

$r_w$: Wake radius
$r_d$: Wake radius immediately behind the turbine
$\alpha$: Entrainment constant
$x$: Axial distance

The entrainment constant ($\alpha$) reflects the speed of wake expansion [33] and is calculated using Equation (6).

$$\alpha = \frac{0.5}{\log\left(\frac{z}{z_o}\right)} \tag{6}$$

where,

$z$: Hub height
$z_o$: Surface roughness considered constant in the current work.

To calculate wake radius '$r_d$' immediately behind the turbine, Equation (7) is used.

$$r_d = r_o \sqrt{\frac{1-a}{1-2a}} \tag{7}$$

where,

$r_0$: Rotor radius

$a$: Axial induction factor

As the wind approaches the wind turbine, it slows down. The ratio of this reduced wind speed and free stream velocity '$u_0$' is axial induction factor '$a$' which is calculated with the help of Thrust Coefficient '$c_t$', as given in Equation (8).

$$c_t = 4a(1 - a) \tag{8}$$

Final equation which provides wake velocity '$u_w$' is as follows,

$$\frac{u_w}{u_0} = \left(1 - \frac{2a}{(1 + \frac{\alpha x}{r_d})^2}\right) \tag{9}$$

where,

$u_w$: Wake velocity

$u_0$: Free velocity

For a single wind turbine, the Jensen wake model simply uses Equation (9). To calculate the wind speed ($u_w$) in a mixed wake for '$N_t$' number of turbines, the Jensen model equates the kinetic energy deficit of a mixed wake with the sum of the kinetic energy deficits of individual turbines.

$$\left(1 - \frac{u_w}{u_0}\right)^2 = \sum_{i=1}^{N_t} \left(1 - \frac{u_i}{u_0}\right)^2$$

$$u_w = u_o \left[1 - \sqrt{\sum_{i=1}^{N_t} \left(1 - \frac{u_i}{u_0}\right)^2}\right] \tag{10}$$

Equations (4)–(10) are incorporated in the work of Jensen and Katic et al. [33,34].

### 2.3. Optimization Process: Genetic Algorithm (GA)

GA mimics reproduction in which parents generate offspring. GA differs from the traditional optimization methods. The fitness function, also known as the objective function, is a key to calculating the power output. Moreover, GA uses convergence criteria to keep the optimization process within an assigned limit. Convergence criteria depend upon a few parameters: population size, mutation, and crossover probability [37].

The following steps are utilized for the wind farm layout optimization using GA, described in Figure 2.

1. Step 1: Initial population size is generated, and design variables are assigned to the wind farm.
2. Step 2: The power output of the initial population is calculated as an objective function.
3. Step 3: Convergence criteria are checked. If it arrives, the optimization process stops; otherwise, move to the next step.
4. Step 4: The selection procedure helps to identify the solutions with the maximum power outputs. Ranking of every solution with variable heights and inter-turbine spacing on the basis of its power output is carried out in the selection process.
5. Step 5: Crossover is the exchange of genes between individuals generated by the GA. Different types of the crossover are commonly used: single-point, two-point, multipoint, and uniform. In the current work, a single-point crossover is used to solve the problem. A cut line is created between both parents, and each parent's first part is combined with the other parent's second part to produce a child.
6. Step 6: Mutation includes the introduction of a gene from outside. A new gene is added randomly to increase the diversity of solutions generated by the crossover. The mutation is done after a certain number of generations. The mutation probability de-

pends on how many times the parts of the chromosomes are mutated. The probability is kept low, as per recommendation.

7.  Step 7: New solutions are generated after the crossover and mutation.
8.  Step 8: New solutions power outputs are calculated by the objective function in step 2. Afterward, the whole optimization process will repeat itself, unless the convergence criteria arrive.

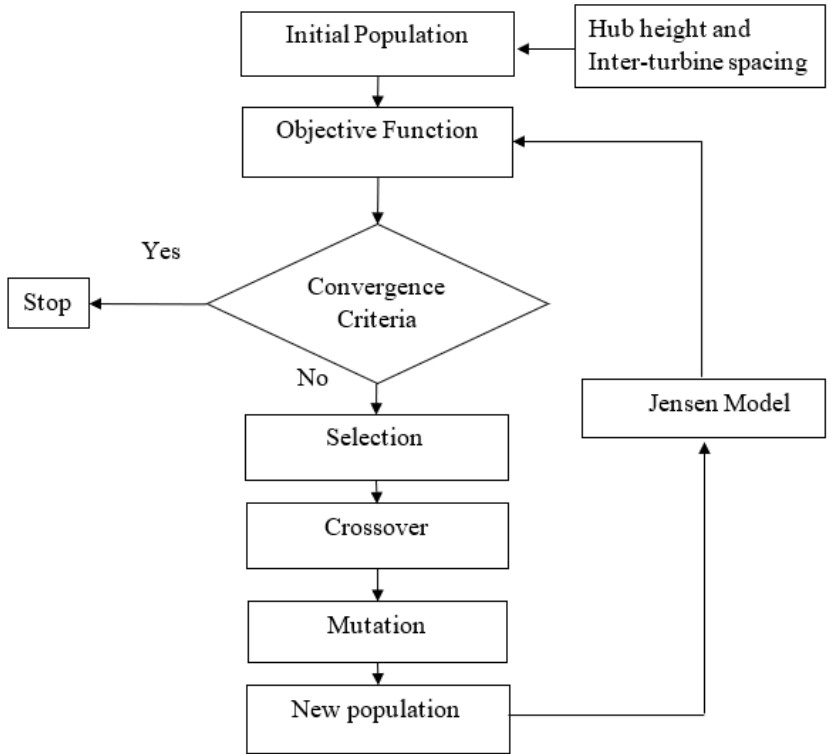

**Figure 2.** Wind farm layout optimization using genetic algorithm.

## 3. Case Study of Jhimpir

### 3.1. Overview

Alternative Energy Board of Pakistan (AEB) has issued a map with the assistance of USAID, as shown in Figure 3 [38]. This map unravels that Pakistan has a huge potential for wind energy generation. Bhutto et al. [39] have done a SWOT analysis of wind energy generation in Pakistan. The research revealed that Pakistan has a huge potential for wind energy generation. In Pakistan, most wind farms are located in Gharo or Sindh Wind Corridor. Wind farms considered in the current work; Second and Third Three Gorges Wind Farms are also installed at Jhimpir in Sindh Wind Corridor. At Jhimpir, wind farms experience wake effects. Feroz et al. [40] proposed Weather Research and Forecasting (WRF) model to forecast wind speed due to wake interference at Jhimpir. Syed et al. [41] an alyzed the impacts of the wake effects of the First Three Gorges Wind Farm and Zorlu Wind Farm on the Fauji Fertilizer Energy Company Limited (FFECL), the wind farm at Jhimpir. The results showed that the wake effects influence the power output of the FFECL wind farm.

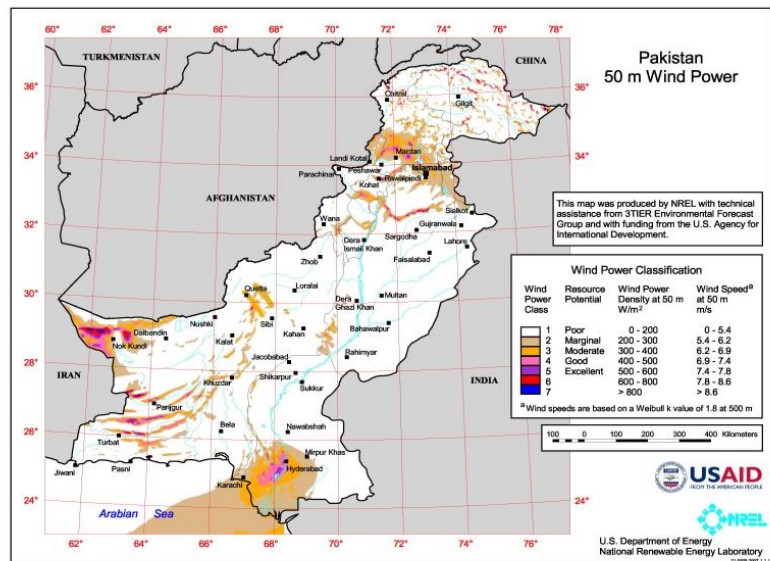

**Figure 3.** Wind resource map of Pakistan [38].

### 3.2. Second and Third Three Gorges Wind Farms (TGWFs) at Jhimpir

Two wind farms that are considered in this work are the second and third Three Gorges Wind Farms (TGWFs) installed at Jhimpir. All of them are operated by Three Gorges Corporation [42]. In Pakistan, many companies have invested a lot of money in the wind sector. This investment has added social and economic capital to Pakistan's society [43]. The specifications of the second and third Three Gorges Wind Farms (TGWFs) analyzed in the current work are listed in Table 1.

**Table 1.** Parameters of the TGWFs [44–46].

| Wind Farm Parameters | Specifications | Units |
|---|---|---|
| Turbine Type | GW82/1500 | NA |
| Hub Height | 85 | m |
| Rotor Diameters | 82 | m |
| Number of turbines | 66 | |
| Rotor Speed | 17 | RPM |
| Average Wind Speed | 7.88 | m/s |
| Cut in Wind Speed | 3 | m/s |
| Survival Wind Speed | 52.5 | m/s |
| Cut Out Wind Speed | 22 | m/s |

### 3.3. Design of the Wind Farm

In Three Gorges Wind Farms (TGWFs), minimum inter-turbine spacing ranges from 330 m to 360 m [44]. In the current work, minimum inter-turbine spacing is kept constant at 340 m or 4D (four times the diameter of the wind turbine). The minimum inter-turbine spacing is within the safety limits. Normally, minimum inter-turbine spacing is kept within the range of 3D to 5D to avoid inter-turbine collision [47].

The entire wind farm has been divided into a grid-like pattern. Every grid has dimensions of 340 m or 4D, as shown in Figure 4. A turbine is installed at the center of the grid. In this way, inter-turbine spacing never recedes below 340 m, the minimum inter-turbine spacing limit.

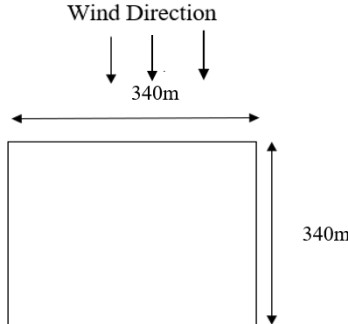

**Figure 4.** Dimension of one grid.

In total, 132 grids are considered in this work, as shown in Figure 5. These grids are presented in 4 rows and 33 columns. The overall dimension of the wind farm is 11,220 m × 1360 m.

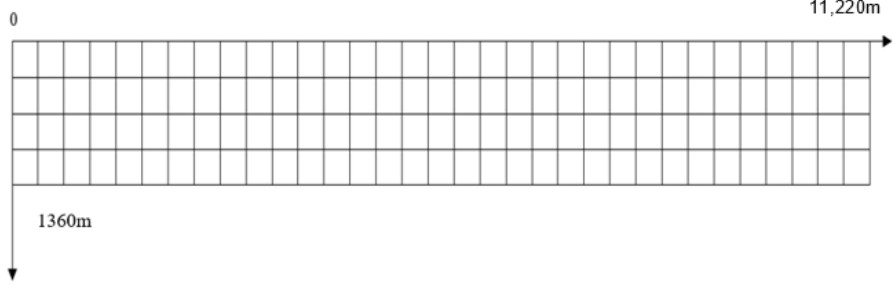

**Figure 5.** A pattern of grids in the TGWFs.

In the current work, wind direction is not varied. The non-variation is present because TGWFs at Jhimpir are oriented southwest. The direction from where the wind flows at Jhimpir is shown in Figure 6.

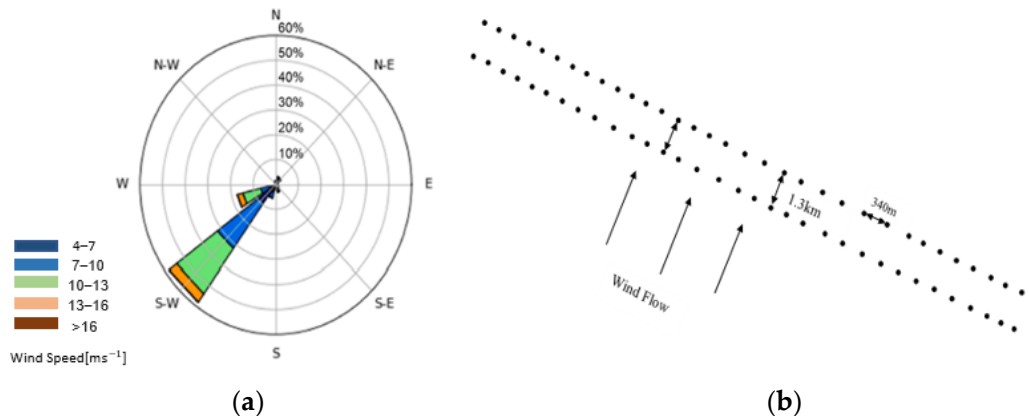

(**a**)          (**b**)

**Figure 6.** Rose Chart and orientation of TGWFs at Jhmpir [41]. (**a**) Rose Chart of the Jhimpir (**b**) Tilted TGWFs in the south–west direction.

## 4. Results and Discussion

### 4.1. Case 1: The Original Three Gorges Wind Farms' (TGWFs) Layout Having the Same Hub Height and Inter-Turbine Spacing without Wake Effect

The original layout of the TGWFs is considered in Case 1. The wake effects are not taken into account. As a result, every turbine faces the free stream velocity. Turbines are installed in the first and the fourth rows shown in Figure 7. As in the original TGWFs, 33 turbines are installed in each of these two rows [48].

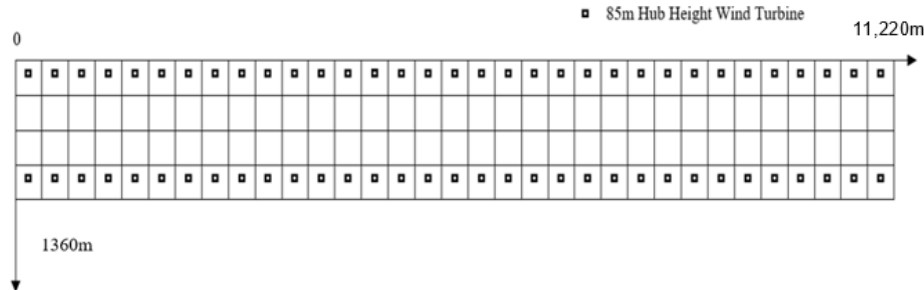

**Figure 7.** The original layout of the TGWFs with the same hub height (25.125, 67.919).

Parameters used in Case 1 to calculate the power output are given in Table 2.

**Table 2.** Parameters to calculate the power output in Case 1.

| Parameters of Case 1 | Values | Units | Symbols |
|---|---|---|---|
| Power Coefficient | 0.48 | NA | $C_p$ |
| Rotor Area | 5281 | $m^2$ | A |
| Density | 1.225 | $kg/m^3$ | P |
| Velocity | 7.88 | $m/s$ | $u_o$ |

The wake effect is not considered in Case 1. Therefore, power calculation is relatively simple. The individual wind turbine power output is calculated using the power Equation (4). As every turbine faces free stream velocity, the power output of the entire wind farm in Case 1 is equal to the power output of the individual wind turbine multiplied by a factor of 66 (total number of turbines at the site) given in Equation (12),

$$p_i = 0.76 \text{ MW} \tag{11}$$

$$p_{total} = p_i \times 66$$
$$p_{total} = 50.16 \text{ MW} \tag{12}$$

50.16 MW is the power output of the TGWFs without wake effect.

### 4.2. Case 2: The Original Three Gorges Wind Farms (TGWFs) Layout Having the Same Hub Height and Inter-Turbine Spacing with Wake Effect

In this case, wake effects are considered while the layout of the wind farm is kept the same as in Case 1, shown in Figure 7. The wake effect in Case 2 expands linearly downwards. It engulfs the downstream wind turbine, as shown in Figure 8.

In Case 2, wind turbines upstream face only the free stream velocity. They have the same power output, as given in Equation (11) (0.76 MW). The total power output of the 33 turbines upstream is calculated by multiplying the power output of individual turbines upstream given in Equation (11) with 33, as shown in Equation (13).

$$p_{upstream} = p_i \times 33$$
$$p_{upstream} = 25.08 \text{ MW} \tag{13}$$

The Jensen wake model is used to calculate the wake velocity for downstream turbines. It relies on several parameters given in Table 3.

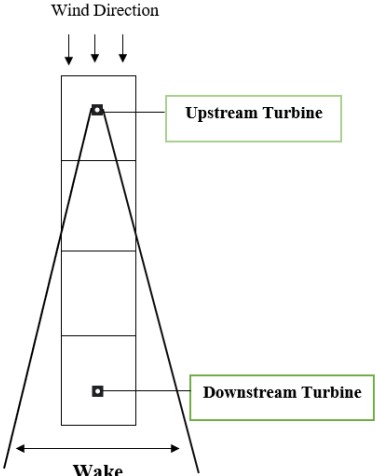

**Figure 8.** Wake effect on the downstream wind turbine.

**Table 3.** Parameters used to calculate the wake velocity in Case 2 [48].

| Parameters of Case 1 | Values | Units | Symbols |
|---|---|---|---|
| Entertainment Constant | 0.21 | NA | $\alpha$ |
| Diameter of Wind Turbine | 82 | m | $D_r$ |
| Axial Distance | 1020 | m | $x$ |
| Axial Induction Factor | 0.28 | | $a$ |
| Wake Velocity | | m/s | $u_w$ |

Downstream wind turbine wake velocity is calculated using the Jensen wake model Equation (14).

$$u_w = u_o \left( 1 - \frac{2a}{(1 + \frac{\alpha x}{r_d})^2} \right) \tag{14}$$

The wake velocity of the downstream wind turbine is 7.71 m/s. The obtained velocity is used to calculate the power output of the downstream wind turbine with the help of the power Equation (4). After calculating power output of individual turbine, the total power output of 33 turbines downstream given in Equation (15) is calculated. The power output is calculated by multiplying individual turbine power output downstream calculated using Equation (4) with 33. It is done because every turbine downstream is the same distance from the upstream turbine. It experiences the same amount of wake from the upstream turbine. As a result, every turbine downstream produces the same power output.

$$p_{downstream} = 23.43 \text{ MW} \tag{15}$$

Total wind farm power output with wake effect in Equation (17) in Case 2 is calculated by adding Equation (13) for the upstream turbines and Equation (15) for the downstream turbines.

$$p_{total} = p_{upstream} + p_{downstream} \tag{16}$$

$$p_{total} = 48.51 \text{ MW} \tag{17}$$

The total power output in Equation (17) is 3.3% lower than the total power output in Equation (12). This shows that the wake effect reduces the power output of TGWFs. The visual representation of power reduction because of the wake effect is given in Figure 9.

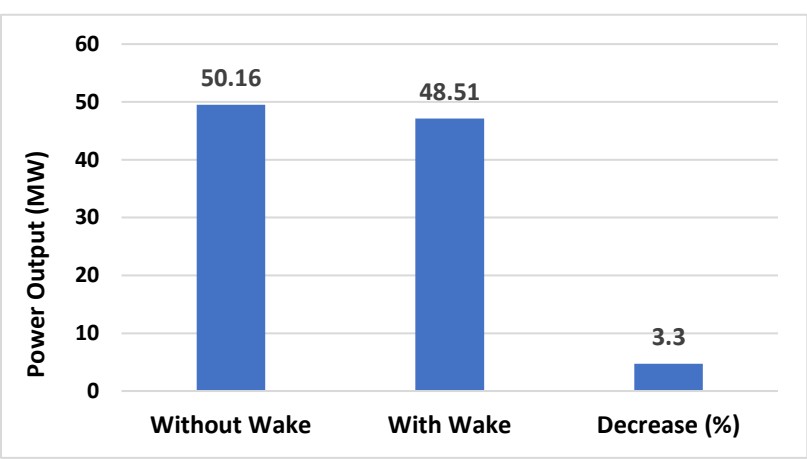

**Figure 9.** Wake impact on the power output of TGWFs.

*4.3. Case 3: Three Gorges Wind Farms (TGWFs) Layout Optimization by Varying Hub Heightsand Inter-Turbine Spacing Using Genetic Algorithm*

In Case 3, layout optimization of TGWFs is performed using the proposed genetic algorithm (GA). Maximizing the total power output of the wind farm is an objective function; theturbine height and inter-turbine spacing are design variables (constraints). The proposed GA is an iterative algorithm that executes a GA code and passes through several iterations. In this work, GA undergoes 1000 iterations (convergence criteria) to arrive at an optimal layout. GA carries out these iterations individually. It stops when iterations reach the limit of the optimization criteria. The solution obtained at the end is an optimal solution. In Case 3, GA optimally installs 66 turbines. This case is divided into two subcases: 3.1 and 3.2.

4.3.1. Subcase 3.1: Optimization of Three Gorges Wind Farms (TGWFs) Layout Having Two Different Hub Heights and Variable Inter-Turbine Spacing with Wake Effect

In subcase 3.1, turbines of two different hub heights are considered. One wind turbine is of higher hub height, and another is of lower hub height. The lower hub height is 85 m, and the higher hub height is 100 m. At lower hub height, the wind speed is 7.88 m/s, and at higher hub height, the wind speed is 8.2 m/s, as given in Table 4 [46]. In general, there are 33 turbines for every hub height.

**Table 4.** Parameters of the wind farm with two different hub heights [48].

| Wind Farm Parameters | Values | Units |
| --- | --- | --- |
| Lower Hub Height | 85 | m |
| Higher Hub Height | 100 | m |
| Wind Speed at Lower Hub Height | 7.88 | m/s |
| Wind Speed at Lower Hub Height | 8.2 | m/s |

The genetic algorithm optimizes the layout of the wind farm with two different hub heights. It installs wind turbines in an optimal layout. It makes sure that the influence of the wake effect remains minimum within the wind farm. In subcase 3.1, the optimal layout of the wind farm generated with the help of the genetic algorithm (GA) is shown in Figure 10.

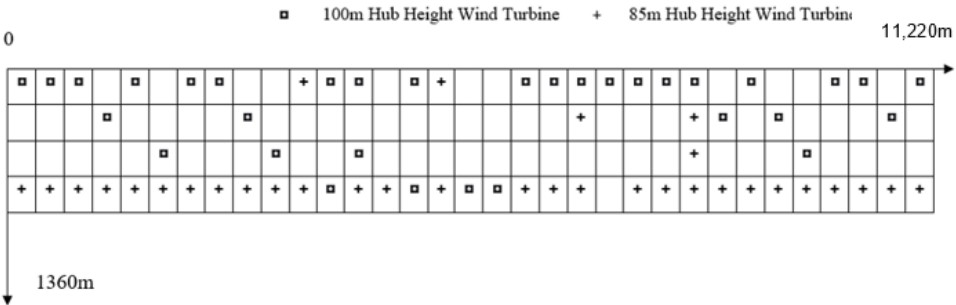

**Figure 10.** Optimal layout of the TGWFs with two different hub heights.

In Figure 10, it is observable that wind turbines remain at the greatest distance from one another. This is because the wake effect gradually withers away as the distance increases.

In subcase 3.1, power output varies with the increase in the number of iterations, as shown in Figure 11. The purpose of every successive iteration is to maximize the power output of the wind farm.

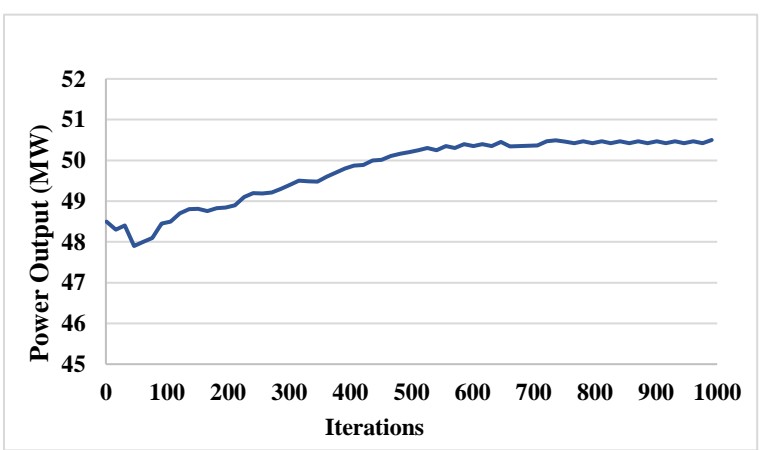

**Figure 11.** Convergence diagram for subcase 3.1.

Table 5 compares wind farms' power outputs for Case 2 with that of subcase 3.1. In Case 2, the wind farm has the same hub height and inter-turbine spacing. The comparison shows that a wind farm in subcase 3.1 produces more power output than the original wind farm with a wake effect in Case 2. This farm in subcase 3.1 generates 50.5 MW of electricity which is 4.1% more than the power output in Case 2.

**Table 5.** Comparison of original and optimized wind farms having two different hub heights.

| Parameters | Original Wind Farm | Optimized Wind Farm |
|---|---|---|
| Number of Turbines | 66 | 66 |
| Total Power Output (MW) | 48.51 | 50.5 |

4.3.2. Subcase 3.2: Optimization of Three Gorges Wind Farms (TGWFs) Layout Having Three Different Hub Heights and Variable Inter-Turbine Spacing with Wake Effect

In subcase 3.2, turbines of three different hub heights were considered, including 80 m, 90 m, and 100 m. Wind speeds at hub heights 80 m, 90 m, and 100 m were assumed to be 7.7 m/s, 7.9 m/s, and 8.2 m/s, respectively, given in Table 6. Out of the total of 66 turbines, 22 were of every hub height.

**Table 6.** Parameters of the wind farm with three different hub heights [48].

| Wind Farm Parameters | Values | Units |
|---|---|---|
| Lower Hub Height | 80 | m |
| Intermediate Hub Height | 90 | m |
| Higher Hub Height | 100 | m |
| Wind Speed at Lower Hub Height | 7.7 | m/s |
| Wind Speed at Intermediate Hub Height | 7.9 | m/s |
| Wind Speed at Lower Hub Height | 8.2 | m/s |

In subcase 3.2, the optimal layout of the TGWFs generated with the help of the genetic algorithm, is shown in Figure 12.

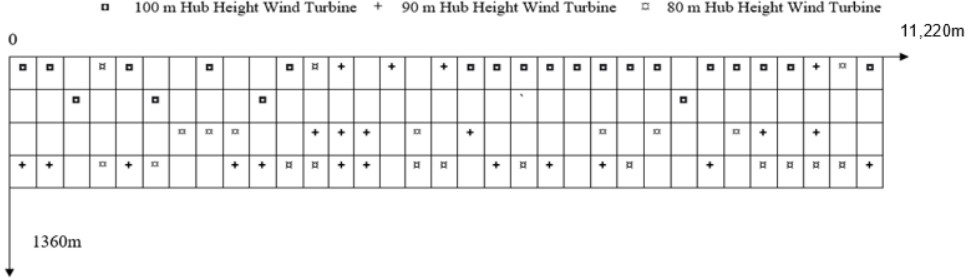

**Figure 12.** Optimal layout of the TGWFs with three hub heights.

Figure 12 shows that the turbines with higher hub heights were mostly in upstream positions, whereas turbines with lower hub heights were mostly in downstream positions. The greater number of higher hub height wind turbines faced the free stream velocity. When the higher hub height turbines were exposed to the free stream velocity, more power output was produced compared to the lower hub height wind turbines. More power output is owing to multiple factors, including the reduced influence of the ground and the greater wind speed at a greater distance from the ground. In subcase 3.2, variation in wind farm power output with the increase in the number of generations is shown in Figure 13.

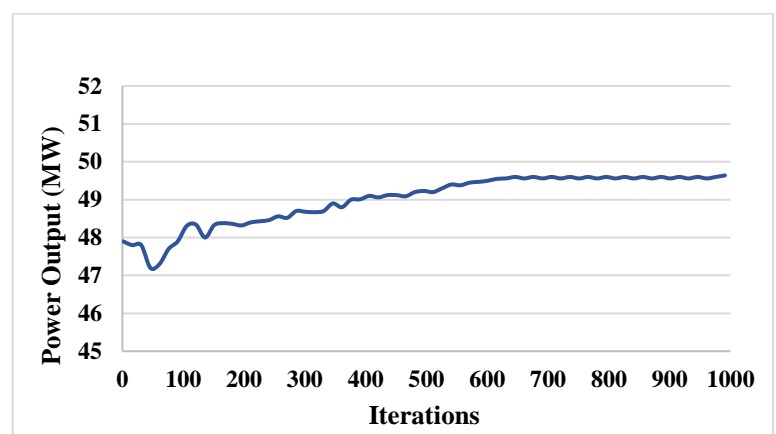

**Figure 13.** Convergence diagram for subcase 3.2.

Table 7 shows a comparison between wind farms power outputs in Case 2 and subcase 3.2. In Cases 1 and 2, hub height and inter-turbine spacing were kept constant, whereas, in subcase 3.2, hub height and inter-turbine spacing are varied. The variation has a large impact on the wind farm power output. As per Table 7, a wind farm in subcase 3.2 produces more power output than the wind farm in Case 2. The wind farm in subcase 3.2 generates 49.64 MW of electricity, which is 2.3% higher than that of 48.51 MW in Case 2.

**Table 7.** Comparison of original and optimized wind farms having three different hub heights.

| Parameters | Original Wind Farm | Optimized Wind Farm |
|---|---|---|
| Number of Turbines | 66 | 66 |
| Total Power Output (MW) | 48.51 | 49.64 |

Table 8 compares the power outputs of all the cases. Overall, Table 8 shows that the wind farm with two different hub heights (elucidated in subcase 3.1) produces more power output than the wind farm with three different hub heights (elucidated in subcase 3.2).

**Table 8.** Comparison of power outputs of the wind farm in three cases.

| Sr. No. | Cases | Description | Power Output (MW) | % Power Increase in Comparison to Case 2 |
|---|---|---|---|---|
| 1 | Case 1 | Original Three Gorges Wind Farms layout without wake effect: Original wind farm layout having the same hub height and inter-turbine spacing while the wake effect is not considered. | 50.16 | |
| 2 | Case 2 | Original Three Gorges Wind Farms layout with wake effect: Original wind farm layout having the same hub height and inter- turbine spacing while wake effect is considered. | 48.51 | |
| 3 | Subcase 3.1 | Optimized Three Gorges Wind Farms layout with two different hub heights: Optimization of wind farm layout having two different hub heights and variable inter-turbine spacing while wake effect is considered. | 50.5 | 4.10% |
| | Subcase 3.2 | Optimized Three Gorges Wind Farms layout with three different hub heights: Optimization of wind farm layout having three different hub heights and variable inter-turbine spacing while wake effect is considered. | 49.64 | 2.30% |

The wind farm with two different hub heights also generates more power output than the wind farm without the wake effect (elucidated in Case 1). Therefore, it is suggested to install a wind farm with two different hub heights at Jhimpir in order to maximize the power input.

**5. Cost Analysis**

The Mossetti et al. [18] cost model has been mostly in the literature for wind farm cost estimation. The model relies only upon wind turbines number ($N$). Other parameters are ignored, and it is estimated that the non-dimensional cost of every turbine remains 1. Later, the Jobs and Economic Development Impacts (JEDI) model was considered for the wind farm cost estimation. The model relies upon other parameters as well [49,50]. The equation of the JEDI Model is given in the equation

$$cost\left(H_{ref}\right) = -0.1539 \times P_r - 0.001 \times N + 2 \times P_r \times N + 0.2504 \qquad (18)$$

where,

$H_{ref}$: Reference hub height

$P_r$: Rated power output

$N$: Total number of turbines.

The JEDI model does not incorporate wind farm cost variation with varying hub heights. Ying et al. [24] have tried to solve the hub height cost estimation problem with Mosetti et al.'s [18] cost model. Assumption is made that cost of every wind turbine, includ-

ing a wind turbine with a different hub height, remains non-dimensional. However, the more practical cost model in terms of varying hub heights is introduced by Abdulrahman et al. [51]. The model is termed a simplified cost model. The model assumes that every 1 m increase in hub height leads to a 1/80 time increase in the wind turbine cost share. The simplified Equation (19) of the simplified cost model is given by Wang et al. [52].

$$cost = cost\left(H_{ref}\right)\left((1 + \frac{1}{200}\left(\sum_{i=1}^{N}(\frac{H_i - H_{ref}}{N})\right)\right) \tag{19}$$

where,

$H_i$: Hub height of the individual wind turbine.

Application of a simplified cost model reveals that the increase in cost with varying hub heights in subcase 3.1 is 3.75% in comparison to Case 2 with the same hub heights. In subcase 3.2, it is 5%. Considering the increase in power output in subcases 3.1 and 3.2, the cost increaseis acceptable. The cost increase investment is a one-off, whereas the power output increase is 4.1% in subcase 3.1 and 2.3% in subcase 3.2, is long-term.

## 6. Conclusions

In this work, a novel method is proposed to maximize the power output of Second and Third Three Gorges Wind Farms (TGWFs) at Jhimpir, Sindh, using a genetic algorithm. Wind farms have lower power outputs due to the wake effects. In the proposed method developed to optimize the wind farm layout, the wake effect was given due consideration and was analyzed using the Jensen wake model. The Jensen wake model was then combined with the genetic algorithm (GA) to optimally install the wind turbines in TGWFs. The optimal layout of TGWFs was achieved while passing through three different cases. The cases were concerned with the wind farm layout optimization: Case 1, the same hub height and inter-turbine spacing without wake effect; Case 2, the same hub height and inter-turbine spacing with wake effect, and Case 3, thevariable hub height and inter-turbine spacing with wake effect.

It has been observed through these three cases that variation in hub heights and inter-turbine spacing increases the power output of TGWFs. In subcase 3.1, the increase in power output of TGWFs is 4.1% compared to the original TGWFs with the wake effect in Case 2. In subcase 3.2, the increase in power output is 2.3% compared to Case 2. TGWFs were also subjected to cost analysis. Varying hub heights of TGWFs in subcase 3.1 increases the cost by almost 3.75% compared to the cost of Case 2 with the same hub heights. In subcase 3.2, the cost increase is about 5%. The cost increase is tolerable. TGWFs have achieved considerably larger power output with a slight increase in initial investment. Hence, future wind farms at Jhimpir, specifically, and in Pakistan, generally, should include variable hub heights and inter-turbine spacings to achieve maximum power output.

Further work is needed to be done on some factors that are not considered in the current work, including the influence of ground and terrain on the TGWFs power output. The intensity of sound in TGWFs is another factor that needs to be analyzed to ensure reduced noise pollution and the comfort of the nearby population.

**Author Contributions:** Conceptualization, M.B.A. and Z.A.; methodology, M.B.A., Z.A. and I.T.; software, M.B.A. and M.R.Y.; verification, M.R.Y. and M.I.; formal analysis, M.B.A. and Z.A.; investigation, I.T., M.R.Y. and S.A.; resources, S.A. and M.I.; writing—original draft preparation, M.B.A., and Z.A.; writing—review and editing, Z.A., S.A. and M.I.; visualization, and S.A.; supervision, Z.A. and M.I. All authors have read and agreed on this version of the manuscript.

**Funding:** Deanship of Scientific Research at King Khalid University, Saudi Arabia, Grant No: R.G.P.2/32/43.

**Data Availability Statement:** Not applicable.

**Acknowledgments:** The authors extend their appreciation to the Deanship of Scientific Research at King Khalid University, Saudi Arabia, for funding this work through the Research Group Program under Grant No: R.G.P.2 /32/43.

**Conflicts of Interest:** The authors declare no conflict of interest.

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
