# Peer review of "A Case Study: Layout Optimization of Three Gorges Wind Farm Pakistan, Using Genetic Algorithm"

_sustainability, doi:10.3390/su142416960_

Round 1
Reviewer 1 Report
This paper public target should be Wind Park (WP) investors who, like me, knows very little of Genetic Algorithms (GA) but understands very well what is needed to make a proper evaluation of a potential WP site. So I think that GA procedures should be more explicit, meaning how it decides what is the evaluation path to follow.
Two GA generated subcases are presented but nothing is said concerning the intermediate layouts leading to these best solutions.
1. Pg1 (34) – “wake effects is the main barrier…”??? maybe rephrase (wind regime, land use, landscape, noise, cost….)
2. Pg3 (118) 2.1 ; pg5 (205) 2.3 – and 2.2? text formatting is also needed.
Pg4 (164) – Eq, (5) it should be “a” and not “a”.
(172) – Eq. (6) should have a reference (original, not [14]).
pg5 (176) – “z0 differs from place to place”, within the WP?
Pg9 (Table 2) – It should be Cp (power coefficient) and nod Betz (constant) limit as the present value includes the machine global efficiency.
(325) – Eq. (11) kW and not MW.
Pg10 (Table 3) – Entrainment constant, a
Axial induction factor, a
Pg11 (Fig. 4.3) – 49.5 MW and 47.1 MW comes from where? It was 50.2 and 48.8 for cases 1 and 2.
Pg12 (Fig. 9) – What about wind directions?
I have a doubt concerning power calculations. It is said (Table 1) that the rated wind speed is 7.88 m/s, so higher wind speeds will not produce more energy than with 7.88 m/s as the rated power (1.5 MW?) was reached. So the power calculations use the WT power curve (with power limits) or just used Eq. (4) regardless the WT characteristics?
Finally, it should be analysed if the cost of increasing WT heights (not only the tower but also the foundations) is compensated with a very few % of increased power (GA vs usual wake model), if the power estimates are correct.
Reviewer 2 Report
Any reference to a practical study for the decrease in the wind turbine power output and efficiency at the downstream (any real case study, not simulation based studies)?? How much reduction in power production takes place due to wake, give some quantitative data (in terms of percentage or any other quantitative units)
Wind speed is intermittent in nature, will the variation in wind speed and hub height complicate the model study of such variations?? (derive equation considering wind speed and hub height variation at the same time) Also the effect of the direction of the flow of wind should also be considered.
Need more discussion of references related to the WAKE effect? Discuss them quantitatively.
Does the wake effect only depend upon the turbine spacing? What could be other factors involved, such as wind direction, turbine height, and wind speed??
No need for reference [10]
Why GA was selected for this study? Why not other better algorithms with possibly better results attainment? Compare with other available algorithms in literature is required?
Reference [20] needs proper formatting.
How the type of turbine and energy conversion system can affect the power output and results?
Define and discuss the constants defined in the article.
Better to present figure 6 with relevance to the actual geographical point of view.
Economical comparison should be included in the case of different hub height wind turbines usage.
With different hub heights, what would be the rotor diameter? Same or different??
Why constant wind speed is assumed?
You can merge figures 10 and 12 for a better comparison of results.
Round 2
Reviewer 1 Report
The (new) text should be carefully revised as there is a number of missing spaces between words all over.
I still I think that GA procedures should be more explicit, meaning how it decides what is the evaluation path to follow.
Pg4 – Eq. (6) should have a reference (original, not [14]). It was not answered. An “entrainment constant reflects the speed of the wake expansion” – sure, but how it depends only on the BL profile?
pg5 (176) – “z0 differs from place to place”, within the WP? It was not answered.
Pg9 (325) – Eq. (11) kW and not MW. Sorry, my error.
Pg10 (Table 3) – It was just a suggestion.
Pg12 (Fig. 9) – What about wind directions? Issue not addressed. I think it is an important matter concerning the WP production.
Reviewer 2 Report
Satisfactory responses to all queries
Author Response
Thank you for your review!